# Parallel Algorithm for Connected-Component Analysis Using CUDA

Dominic Windisch [1,*], Christian Kaever [2], Guido Juckeland [2] and André Bieberle [2]

1. Institute of Power Engineering, Technische Universität Dresden, 01062 Dresden, Germany
2. Helmholtz-Zentrum Dresden—Rossendorf, Bautzner Landstr. 400, 01328 Dresden, Germany
* Correspondence: d.windisch@hzdr.de

**Abstract:** In this article, we introduce a parallel algorithm for connected-component analysis (CCA) on GPUs which drastically reduces the volume of data to transfer from GPU to the host. CCA algorithms targeting GPUs typically store the extracted features in arrays large enough to potentially hold the maximum possible number of objects for the given image size. Transferring these large arrays to the host requires large portions of the overall execution time. Therefore, we propose an algorithm which uses a CUDA kernel to merge trees of connected component feature structs. During the tree merging, various connected-component properties, such as total area, centroid and bounding box, are extracted and accumulated. The tree structure then enables us to only transfer features of valid objects to the host for further processing or storing. Our benchmarks show that this implementation significantly reduces memory transfer volume for processing results on the host whilst maintaining similar performance to state-of-the-art CCA algorithms.

**Keywords:** connected-component analysis; image stream processing; parallel computing, CUDA

## 1. Introduction

Connected-component analysis (CCA) is a common problem in many imaging applications, such as medical imaging [1,2], automated quality assurance [3,4] or real-time automotive applications [5,6]. A novel application is image-based process control using ultrafast electron beam X-ray computed tomography (UFXCT) [7]. UFXCT is a non-invasive imaging technique yielding up to 8000 images per second. This image stream is reconstructed and processed in real-time on a GPU using the so-called Real-time Image Stream Algorithms (RISA, [8,9]) framework. CCA shall now be used in RISA to extract object features from this reconstructed image stream. Based on these extracted features, the UFXCT scanning position is adapted during scanning to track moving objects, e.g., a rising gas bubble in a liquid column. For stable tracking, features must be extracted within a latency of at most 5 ms [10]. Consequently, the connected-component analysis in RISA must also be performed in an efficient and computationally cheap manner on the GPU. After the analysis, resulting features must also be transferred to the host for long-term storage and possible further analysis. Lastly, the connected-component analysis should be implemented as a separate processing step independent of the labeling to maintain maximal flexibility, e.g., when pre-processing of the labels is required. One such example is the aforementioned analysis for UFXCT which requires persistent labels for the same logical object between consecutive frames. Note, however, that RISA can be applied for general purpose image stream processing and is not limited to UFXCT. However, we will use UFXCT as an example application for introducing the design of our CCA algorithm because it imposes the most strict constraints (latency, label persistence).

Given these constraints, CCA for UFXCT can be divided into the steps shown in Figure 1:

(1) Image binarization (e.g., using a global threshold);

(2)   Connected-component labeling (e.g., using the algorithm developed by Playne [11]);
(3)   Label inheritance [12] for objects which are visible in multiple consecutive frames;
(4)   Feature extraction with our proposed tree merge kernel (see Section 3.2);
(5)   Transfer of these features to host memory for storing and further processing.

In this work, we focus on the connected-component analysis. Related work is described in the following Section 2.

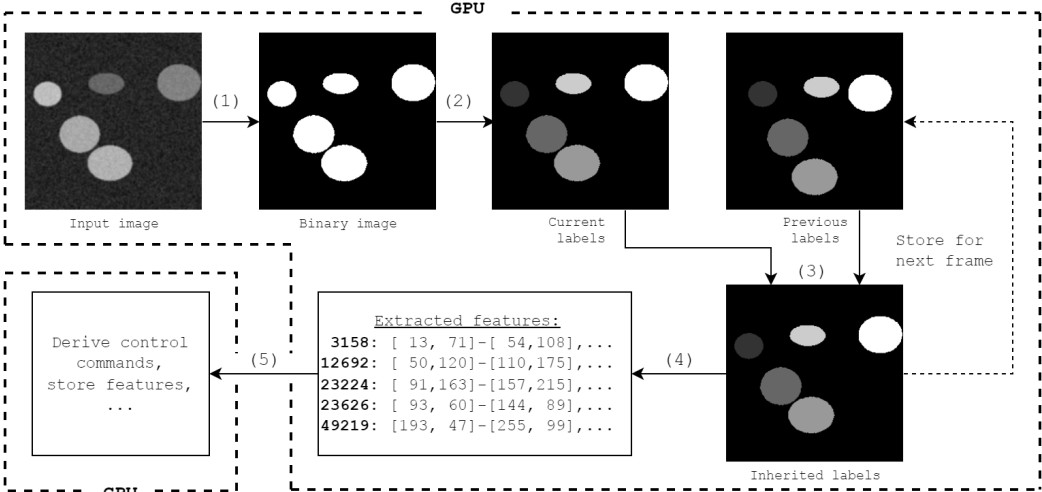

**Figure 1.** Data processing steps for connected component analysis to realize object tracking using UFXCT: (1) binarization, (2) initial connected-component labeling, (3) label inheritance, (4) connected-component analysis, and (5) feature analysis on CPU.

## 2. State of the Art and Related Work

In the context of this paper, we define the connected-component analysis problem such that connected regions of foreground pixels in a binary image ("true"), representing logical objects, are labeled identically. For each of these objects, a set of properties, like overall area, centroid position and bounding box, is to be extracted. To solve those problems, many algorithms were already developed which can be coarsely classified into sequential (e.g., [13–15]) and parallel algorithms (e.g., [16–19]). In this paper, we will focus on parallel implementations of connected-component analysis (CCA) for execution on NVIDIA GPUs. Note that many approaches for CCA use hybrid kernels which label and analyse the components in a single step (e.g., [16,18]). This approach is not applicable in our case due to the inheritance constraint. However, these algorithms can typically also be applied to a labeled image which allows us to compare their performance with our own implementation.

A major issue for any parallel CCA implementation is that multiple independent processes may try to update the same component's features concurrently. Resolving this typically involves using atomic instructions which serialize any memory accesses to maintain a well-defined state. However, serializing the process counteracts the initial idea of increasing performance by executing operations in parallel when using GPUs.

To circumvent the serialization, different approaches were already developed. HA4 [16] is an algorithm especially designed to circumvent slow-downs due to ineffective hardware usage (e.g., non-optimal memory accesses). It uses a single thread per image row which traverses the image column-wise. Labels and features can be updated in the same iteration using specific warp-level primitives yielding very good performance. Similarly, [19] further reduced possible causes for serialization due to merging conflicts between threads. This is achieved by using on-the-fly feature merges or conflict detection, again, implemented with low-level CUDA functions.

With these performance improvements, transfer time from device to host increasingly dominates overall execution time when feature processing on the CPU is required. The

CCA approaches introduced above do not consider reduction of transfer volume. Instead, a set of arrays (one per feature) is filled by the CCA algorithms typically yielding sparsely populated arrays. Naïvely transferring these to the host leads to longer transfer durations and increased computing costs on the CPU to iterate and process a large set of empty or invalid array entries.

In our contribution, we introduce CCA based on a tree merging algorithm which creates a tree structure whose root node contains the size of the total tree. Hence only valid property blobs are downloaded from the device to host memory drastically reducing the transfer volume for most use-cases.

## 3. Materials and Methods

### 3.1. Boundary Conditions for CCA in Image Stream Processing

The proposed algorithm was developed with these design considerations stemming from its application in image stream processing:

- Image contents between consecutive images are assumed to contain approximately the same number of objects.
- Object features need to be transferred to host memory for storing or further processing.
- The data processing framework (RISA) may require large portions of the available memory bandwidth for copy operations between host and device memory. The CCA algorithm should therefore use minimal memory bandwidth.
- Applications may require pre-processing of labels before the feature extraction. Hence, CCA must be performed as a separate step and not during the initial labeling process.

### 3.2. Tree Merge Kernel

Consequently, the proposed Tree Merge Kernel (TMK) consists of these steps:

1. Initialization;
2a. Tree merging (in shared memory);
2b. Tree merging (in global memory);
3. Updating tree size information;
4. Tree balancing.

Given a labaled image, the developed algorithm computes a binary tree of object feature structs (`OFS`) which are stored in a contiguous array (`F`). Each `OFS` includes information about the object itself (label, number of pixels, boundary box, sum of coordinates x, sum of coordinates y) as well as the tree structure (parent index, left and right child indices, total number of child nodes). Each `OFS` can therefore be interpreted as a tree node. After balancing the tree, all the valid nodes are moved to the beginning of the array. This allows us to transfer only the valid nodes at the beginning of the array to the host in a single copy operation, which we assume to lower required memory transfer bandwidth.

The steps to build this tree structure are detailed below. For the code shown, we define these variables: `F` = array of object feature structs, `INVALID_LABEL` = constant value indicating a pixel is not a foreground pixel, `L` = array of image size containing one label per pixel, `tig` = thread index in the grid used when calling the kernel.

#### 3.2.1. Initialization

The input to the algorithm is a labeled image of size $width \cdot height = numPixels$ pixels. This image can contain at most $numPixels/2 = maxNumObjects$ distinct objects (i.e. a chessboard pattern) because two neighbouring foreground pixels can not be part of different objects. If they were touching, they would be considered the same object. Hence, we can initialize the feature extraction allocating an array `F` of size $maxNumObjects$ `OFS`s. In the initialization kernel, we use $maxNumObjects$ threads which initialize `F` by processing two neighbouring pixels as shown in Algorithm 1.

---

**Algorithm 1:** initialize(F, L, width)

---

1　$i_0 \leftarrow 2 \cdot \texttt{tig}, i_1 \leftarrow 2 \cdot \texttt{tig} + 1$
2　**for** $k \leftarrow 0$ **to** 1 **do**
3　　　$x_k \leftarrow \text{mod}(i_k, \text{width}), y_k \leftarrow i_k / \text{width}$
4　　　$\text{is\_valid}_k \leftarrow (L[i_k] \neq \texttt{INVALID\_LABEL})$
5　**if** $!\text{is\_valid}_0$ AND $!\text{is\_valid}_1$ **then**
6　　　$F[\texttt{tig}].\text{label} \leftarrow \texttt{INVALID\_LABEL}$
7　　　**return**
8　$F[\texttt{tig}].\text{sum}_x \leftarrow x_0 \cdot \text{is\_valid}_0 + x_1 \cdot \text{is\_valid}_1$
9　$F[\texttt{tig}].\text{sum}_y \leftarrow y_0 \cdot \text{is\_valid}_0 + y_1 \cdot \text{is\_valid}_1$
10　$F[\texttt{tig}].\text{count} \leftarrow \text{is\_valid}_0 + \text{is\_valid}_1$
11　**if** $\text{is\_valid}_0$ AND $\text{is\_valid}_1$ **then**
12　　　$F[\texttt{tig}].\text{label} \leftarrow L[i_0]$
13　　　$F[\texttt{tig}].x_{min} \leftarrow \min(x_0, x_1), F[\texttt{tig}].x_{max} \leftarrow \max(x_0, x_1)$
14　　　$F[\texttt{tig}].y_{min} \leftarrow \min(y_0, y_1), F[\texttt{tig}].y_{max} \leftarrow \max(y_0, y_1)$
15　**else**
16　　　$F[\texttt{tig}].\text{label} \leftarrow L[i_0] \cdot \text{is\_valid}_0 + L[i_1] \cdot \text{is\_valid}_1$
17　　　$F[\texttt{tig}].x_{min} \leftarrow x_0 \cdot \text{is\_valid}_0 + x_1 \cdot \text{is\_valid}_1$
18　　　$F[\texttt{tig}].x_{max} \leftarrow x_0 \cdot \text{is\_valid}_0 + x_1 \cdot \text{is\_valid}_1$
19　　　$F[\texttt{tig}].y_{min} \leftarrow y_0 \cdot \text{is\_valid}_0 + y_1 \cdot \text{is\_valid}_1$
20　　　$F[\texttt{tig}].y_{max} \leftarrow y_0 \cdot \text{is\_valid}_0 + y_1 \cdot \text{is\_valid}_1$

---

F then contains *maxNumObjects* OFSs which are initially disjointed tree nodes. Each node either holds a valid tree (when at least one of the associated pixels is a foreground pixel) or an invalid tree (when none of the associated pixels are foreground pixels).

### 3.2.2. Tree Merging

These initially independent trees are then iteratively merged with doubling granularity between iterations. Trees are merged in shared memory first. If the granularity exceeds the shared memory capacity, the resulting trees are then finally merged in global memory. As shown in Figure 2, each active thread (boxes with bold outline) handles two tree nodes per iteration in one of four scenarios (colored boxes):

(a)　The source node is invalid, i.e., empty. The target node therefore remains unchanged and the assigned thread returns.

(b)　The source and target node contain different labels. The source node is inserted as a child node into the target tree (as indicated by the dashed arrow). Note that this insertion is performed using `atomicCAS` to prevent multiple threads from claiming the same empty spot in the target tree.

(c)　The target node is invalid and therefore replaced by the valid source node. The source node's children are updated with the new parent address.

(d)　The source and target node are labeled equally and are merged by updating their features. Because a label cannot occur twice in an existing tree there are no merging conflicts, hence feature updates (e.g., min/max of coordinates) are calculated nonatomically. Existing pointers to child nodes are removed to prevent conflicts. These cut-off child nodes are inserted back into the tree by their respective thread in subsequent iterations.

Between the iterations, the threads need to be synchronized using `__syncthreads` when working on block level and `cooperative_groups::this_grid().sync()` when working on global scope.

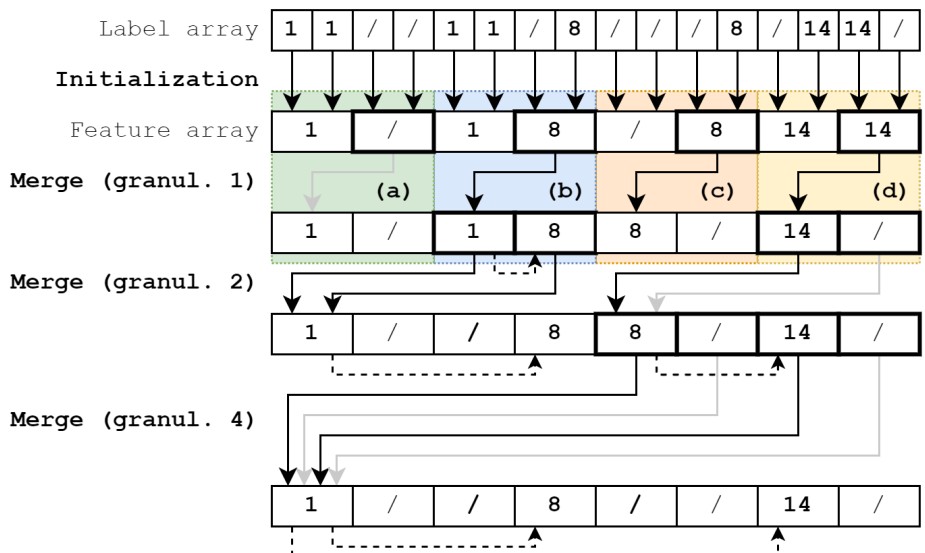

**Figure 2.** Iterative merging of initially disjoint trees on GPU. Boldly outlined boxes represent source nodes for each step (each assigned a separate thread). Note that threads for invalid source nodes return early (indicated by gray arrows). Dashed arrows indicate parent-child relationships between nodes. Scenarios (a) through (d) are described in Section 3.2.2.

### 3.2.3. Updating Tree Size Information

After building the tree, the size of each node, i.e., the total number of respective child nodes, is calculated. Updating the size information during building the tree would require propagating size increments atomically multiple times. Instead, we are opting to update the tree sizes after building it by assigning one thread per leaf node. This way, the minimum number of threads (one per object) perform the size updates reducing atomic instructions. As shown in Algorithm 2, one thread per leaf node traverses the tree up until the root whilst updating the size of each visited node. If the visited node has two child nodes, the size update is performed atomically using `atomicCAS`. When only one child node is present, the update is performed non-atomically.

---

**Algorithm 2:** updateTreeSize(F, maxNumObjects)

---

1 **if** `tig` $\geq$ maxNumObjects **then return**
2 $i_{curr} \leftarrow$ `tig`
3 **if** (!F$[i_{curr}].isValid()$) OR (!F$[i_{curr}].isLeaf()$) **then return**
4 F$[i_{curr}]$.size $\leftarrow 1$
5 **while** F$[i_{curr}]$ has parent node **do**
6      $i_{prev} \leftarrow i_{curr}$
7      $i_{curr} \leftarrow$ F$[i_{curr}]$.parentNodeIndex
8      hasBothChildren = F$[i_{curr}]$.hasLeftChild() AND F$[i_{curr}]$.hasRightChild()
9      **if** hasBothChildren **then**
10          result = `atomicCAS`(&F$[i_{curr}]$.size, 0, F$[i_{prev}]$.size + 1)
11          **if** result != 0 **then** F$[i_{curr}]$.size $\leftarrow$ result + F$[i_{prev}]$.size
12      **else**
13          F$[i_{curr}]$.size $\leftarrow$ F$[i_{prev}]$.size + 1

---

### 3.2.4. Tree Balancing

After updating the tree sizes, the valid tree resides in the sparsely populated array F. To maximally shrink the data volume before transfer to host, all valid nodes are shifted to the beginning of the array essentially balancing the tree. With the total number of valid

nodes known in the root node, the host can now copy only the valid nodes located at the beginning of the array. The algorithm for this balancing is shown in Algorithm 3.

---

**Algorithm 3:** balance(F)

---
1   f ← F[tig]
2   **if** f.isValid **then**
3      level ← 0, $i_{curr}$ ← 0, $i_{final}$ ← 0
4      **while** true **do**
5          **if** F[$i_{curr}$].label == f.label **then**
6              **break**
7          goLeft ← f.label & (1 « depthInTree)
8          numElemsLeftSubtree = F[F[$i_{curr}$].leftChildIdx].size()
9          $i_{final}$ += 1 + (!goLeft · numElemsLeftSubtree)
10         $i_{curr}$ = goLeft ? F[$i_{curr}$].leftChildIdx : F[$i_{curr}$].rightChildIdx
11   cooperative_groups::this_grid().sync()
12   **if** f.isValid **then** F[$i_{final}$] ← f

---

Note that the information about the total number of nodes resides in GPU memory. To copy the correct amount of nodes, i.e., at least all valid nodes, it is assumed that the number of objects does not change drastically between consecutive frames. Initially, a user-provided number of objects is downloaded. It can then be checked if all valid nodes were copied by accessing the size of the root node in host memory. If not all nodes were copied, the remaining nodes are transferred in a second copy operation. In consecutive frames, the number of nodes in the previous frame (plus a safety margin) is copied in the first copy operation. If the assumption of roughly constant number of objects holds true, this will eliminate the necessity for the second copy operation whilst also only copying a fraction of the feature array F.

### 3.3. Performance Evaluation

We benchmarked the performance by executing a processing pipeline in RISA consisting of three stages:

- A "loader stage", which loads the respective part of the benchmark data set into device memory and sends it out in fixed intervals;
- A "connected-component analysis stage", which consumes the data stream from the benchmark loader and performs CCA (including labeling using Playne), and finally forwards the labeled images for further processing;
- A "discarder stage", which removes images from the pipeline.

For a comparison to a state-of-the-art CCA algorithm, we also implemented the hardware accelerated 4-connectivity algorithm in the CCA-stage as outlined in [16]. We used the YACCLAB data set [20] for our tests. This dataset contains a set of benchmark images from different applications which have been used for benchmarking CCA algorithms in different studies. In particular, we used the "random/granularity" and "random/classical" data sets (see example images in Figure 3). The benchmarks were run on the two systems described in Table 1.

Each benchmark was run by reading the set of ten images of the particular parameters (e.g., density 50%, granularity 4) into the device memory. This set is streamed through the processing stage repeatedly until 1000 images are processed in total. Using 'nsys', we assessed the kernel execution and memory transfer (device to host) durations.

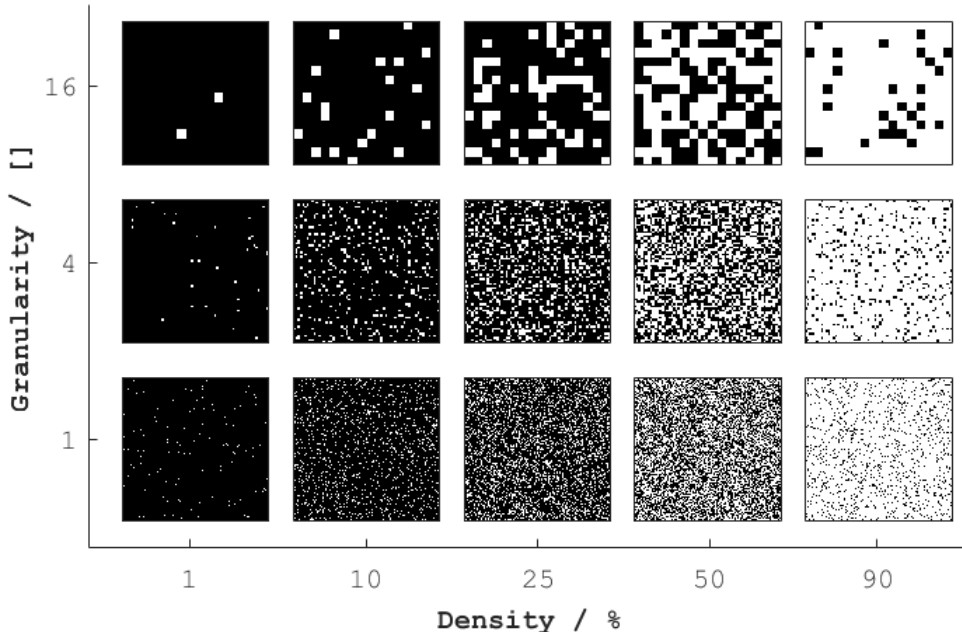

**Figure 3.** Examples of the used images from the YACCLAB dataset. White pixels are considered foreground pixels.

**Table 1.** Benchmark system specifications.

| Component | Specification | |
| | System A | System B |
| --- | --- | --- |
| CPU | AMD EPYC 7343 (16 × 3.9 GHz) | Intel Xeon E5-2637 (16 × 3.5 GHz) |
| GPU | NVIDIA RTX 3080 (10 GB VRAM) | NVIDIA GTX 1080 (8 GB VRAM) |
| Mainboard | Supermicro H12SSL-i | Fujitsu D3358-A1 |
| RAM | 4 × 32 GB Samsung M393A4K40DB3-CWE (DDR4, 3200 MHz) | 8 × 16 GB Samsung M393A2G40DB0-CPB (DDR4, 2133 MHz) |
| OS | Ubuntu 20.04 (5.15.0-46-generic) | Ubuntu 20.04 (5.4.0-131-generic) |
| GCC | 10.3.0 | 10.1.0 |
| NVCC | 11.6 | 11.2 |
| CUDA driver | 510.47.03 | 460.73.01 |

## 4. Results

Figure 4 shows the execution times on the System A for the proposed tree merge kernel (TMK) compared to the HA4 algorithm. Benchmarks on the System B showed qualitatively similar results and are not shown. Note that the execution time for TMK includes generation and balancing of the tree, whereas the HA4 algorithm does not create a tree structure. Instead, the features array F is filled and transferred completely. The results show that the increased computational effort on the device, i.e., building and balancing the tree, yields an overall lower execution time due to the drastically reduced memory transfer time in most cases.

Further, it can be seen that the tree merge kernels execution time scales with the total number of connected components rather than the number of foreground pixels, i.e., density. Hence, it performs worst for densities around 25% which yield the largest number of objects in the random images. Contrary, the tree merge algorithm remains unaffected by the percolation threshold around 60% which typically poses a bottleneck for CCA algorithms like HA4. This slowdown for algorithms like HA4 is a result of the sudden increase in object sizes at the percolation threshold. This leads to more conflicts between threads voting on

the objects properties which need to be serialized. TMK, on the other hand, merges object properties non-atomically (as explained in Section 3.2.2(d)). Hence, it is not negatively effected by the increase in object size. Instead, TMK needs to perform atomic operations, i.e., potential serialization, when building the tree structure which becomes more expensive for larger numbers of objects.

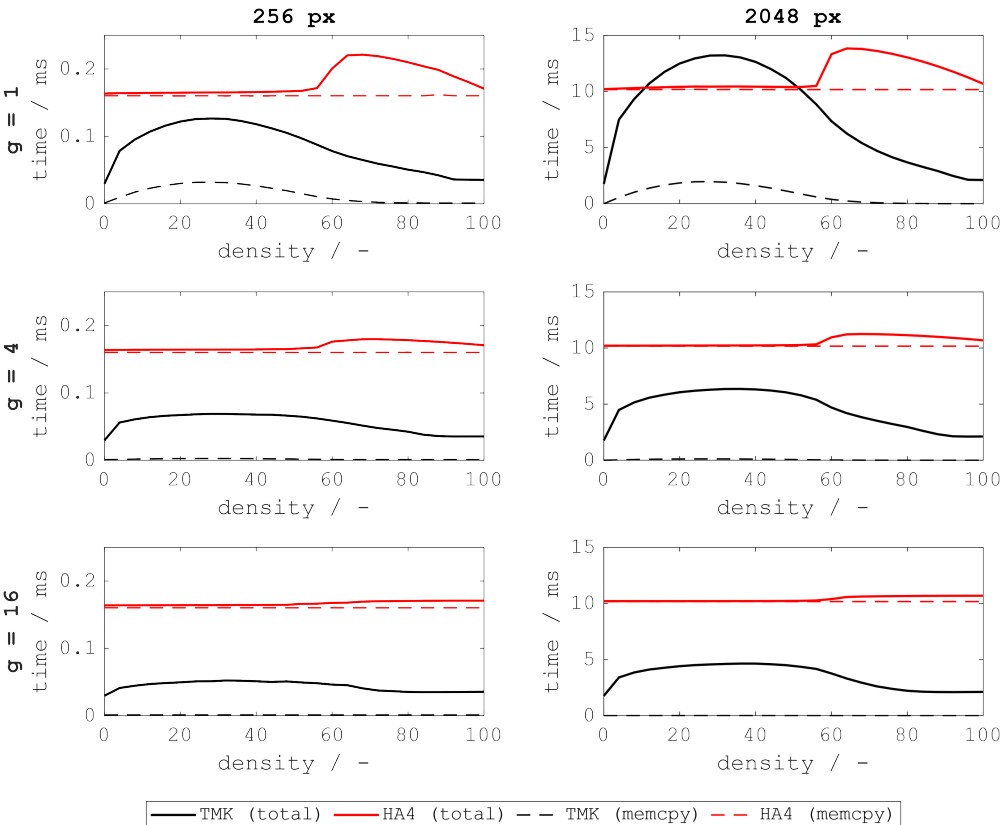

**Figure 4.** Comparison of execution time (System A) vs. density for different granularities g for images with 256 × 256 pixels and 2048 × 2048 pixels. Solid lines inidicate total execution time (including `cudaMemcpy`). Dashed lines show the respective `cudaMemcpy` time for the feature array (device to host). Errobars are omitted for clarity (standard deviation < 3% in all cases).

Regarding varying image sizes, the results reveal TMKs different effectiveness for small, medium and large image size. The average execution time with respect to the image size is shown in Figure 5. The portions of total execution time for each part of TMK are shown in Figure 6.

For small image sizes (≤128 px), the overhead for kernel launches is significant. Even though much of the tree merging is performed in shared memory, the overall computational effort outweighs the possible savings in memory transfer time.

For medium image sizes (256 px to 1024 px), the TMK performs best due an increase in possible memory transfer savings whilst processing large portions of the tree merging in shared memory.

For large image sizes (≥2048 px), much of the tree merging must be performed using global memory. This significantly slows down execution. In total, the additional computation time required for the global memory merging outweighs the increased savings in memory transfer time.

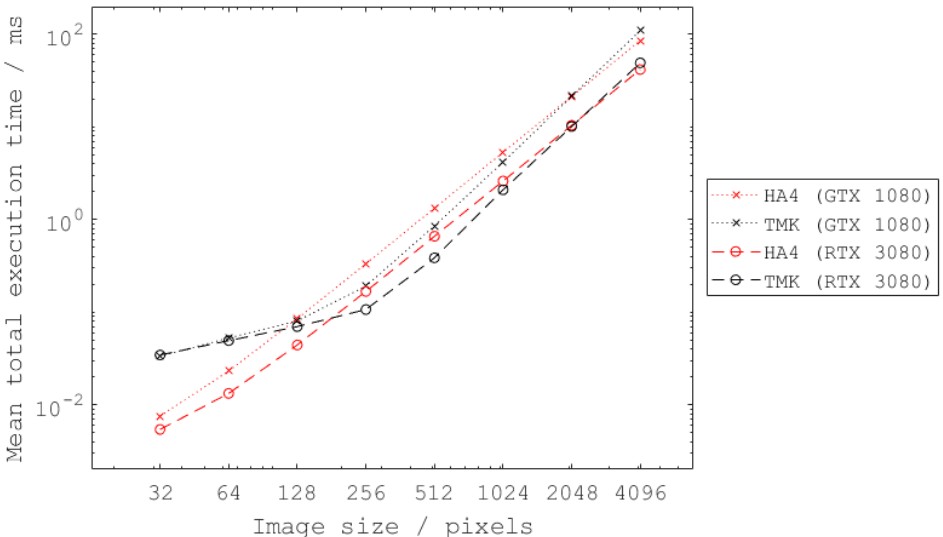

**Figure 5.** Mean total execution time for different image sizes. Mean is computed by averaging the total execution time for images with densities from 10% to 90% and granularity 1. Errorbars are omitted for clarity (standard deviation < 3% in all cases).

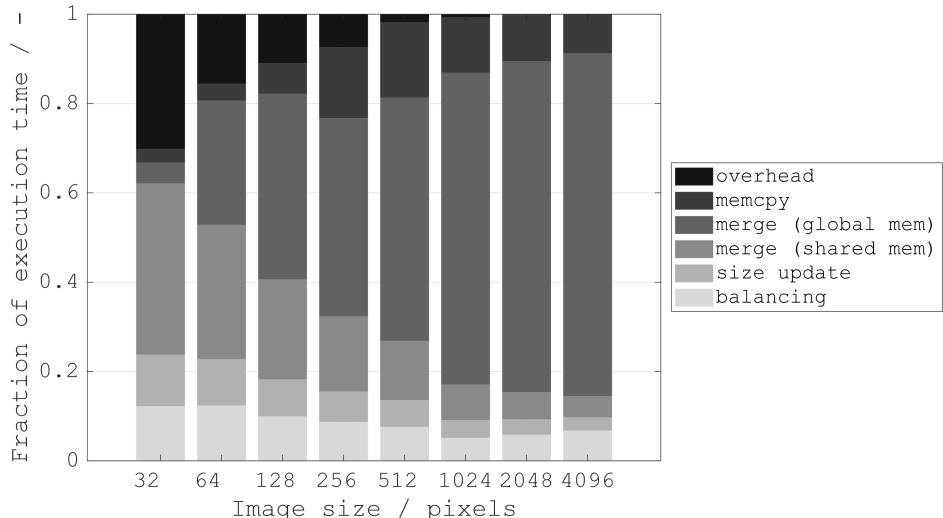

**Figure 6.** Fraction of kernel execution time for different image resolutions. Respective execution times are extracted using `nsys`. Overhead is calculated as remaining time between the sum of kernel execution times and total execution time, i.e., from start of the first kernel to the end of the final `memcpy`.

## 5. Discussion

Our investigations have shown that the proposed tree merge kernel (TMK) reduces overall execution times for connected component analysis when taking into account transfer times between device and host. It performs best for medium image sizes (256 px to 1024 px). For larger images, total execution time is similar to HA4 due to TMKs increasing dependency on global memory accesses. Still, required memory bandwith is drastically reduced when compared to naïvely transferring the (typically sparse) object features array from the device. This advantage is even more pronounced for images with larger granularity, i.e., less objects in total. Contrary, mere feature calculation using TMK is slower than HA4 for most cases, with noteworthy exceptions for image densities around the percolation threshold.

For future work, it may be worthwhile combining the advantages of both approaches in an algorithm which can quickly perform CCA (e.g., HA4) followed by compression of the features array before transfer to the host.

**Author Contributions:** Conceptualization, D.W. and C.K.; methodology, D.W. and C.K.; software, D.W. and C.K.; validation, D.W. and C.K.; formal analysis, D.W.; investigation, D.W.; resources, G.J. and A.B.; data curation, D.W.; writing—original draft preparation, D.W.; writing—review and editing, A.B. and G.J.; visualization, D.W.; supervision, A.B.; project administration, D.W. and A.B.; funding acquisition, A.B. and G.J. All authors have read and agreed to the published version of the manuscript.

**Funding:** This research was funded by Deutsche Forschungsgemeinschaft under grant number HA 3088/26-1.

**Data Availability Statement:** Publicly available software and data was analyzed in this study. The source code can be found here: https://codebase.helmholtz.cloud/risa (accessed on 27 January 2023). The dataset used for the benchmarks ("YACCLAB") can be found here: https://github.com/prittt/YACCLAB (accessed on 6 October 2022). The subset "random/granularity" was used for Figure 4. For analysis of 256 px images, only the top-left 256 × 256 px region was used. The subset "random/classical" was used for Figures 5 and 6.

**Conflicts of Interest:** The authors declare no conflict of interest.

## Abbreviations

The following abbreviations are used in this manuscript:

| | |
|---|---|
| CCA | Connected-Component Analysis |
| CUDA | Compute Unified Device Architecture |
| FPS | Frames Per Second |
| GPU | Grapical Processing Unit |
| HA4 | Hardware Accelerated 4-connected |
| OFS | Object Feature Struct |
| RISA | Real-time Image Stream Algorithms |
| TMK | Tree Merge Kernel |
| UFXCT | Ultrafast electron beam X-ray Computed Tomography |

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
