# Peer review of "Parallel Algorithm for Connected-Component Analysis Using CUDA"

_algorithms, doi:10.3390/a16020080_

Round 1

Reviewer 1 Report

The authors proposed a parallel algorithm for connected-component analysis (CCA) on GPUs, which is able to reduce the data volume movements from GPU to host. This result was achieved using a tree merging algorithm that assigns the features of the total tree only to the root node. The performance results were compared with another CUDA algorithm, named HA4 [ref n. 16], a CC labeling and analysis algorithms, using the online available dataset YACCLAB  [ref n.20]. They presented several tests where the proposed strategy outperforms HA4 in terms of overall execution time. The proposed algorithm named T(ree)M(erge)K(ernel) is implemented in CUDA C/C++ code made available online.

Several typos and grammar mistakes appear in the manuscript and should be corrected. For example:

  • “YACCLAB data set” should be “YACCLLAB dataset”

  • page 6, line 184: “where” should be “were”

  • page 6, line 188: “memory operation duration” requires a more formal definition, e.g., “memory operations” 

  • etc. 

Please, read and carefully check the overall manuscript.

Moreover, a short description of the YACCLAB could be appropriate, and the manuscript could include some figures among the ones used in the tests.

Apart from the above mistakes and suggestions, the manuscript appears well-written and organized, and  I recommend taking the paper after the above minor revisions.

Author Response

Several typos and grammar mistakes appear in the manuscript and should be corrected. For example:

  • “YACCLAB data set” should be “YACCLLAB dataset”

  • page 6, line 184: “where” should be “were”

  • page 6, line 188: “memory operation duration” requires a more formal definition, e.g., “memory operations”

  • etc.

Please, read and carefully check the overall manuscript.

Thank you for the remarks. We have checked the manuscript and fixed it.

Moreover, a short description of the YACCLAB could be appropriate, and the manuscript could include some figures among the ones used in the tests.

Thank you for the suggestion. We have added examplary images used in the benchmarks (see Figure 3) and extended the description (see bold text on page 6).

Reviewer 2 Report

The articles tackles the well-known trade-off between computation time and communication time (memory access time) in the special case of a concrete algorithm (connected component analysis) that run in parallel in a GPU and needs to transfer its final result to the CPU.

The results obtained by the authors show that their trading of more computation and less communication eventually achieves, on a significant benchmark, a smaller overall runtime for the specific problem on their specific machine, with the exception of the larger problem instances.

It would have be nice to see some theoretical runtime model, that on the one hand side predicts/models the actual runtime on the machine,  and in the other hand side gives more insight into the properties both of the algorithm and of the system, such that the experiments could be exported to other applications (both in  the context of algorithms as well as in the context of using a specific GPU/CPU architecture).

A simple extension could be: just use some different GPU (and/or CPU) and analyse what happens there.

What are the reasons for the fact that HA4 is affected at 60% (percolation threshold) but TMK not? Well, actually it is affected looking at Fig.3, but in a positive sense!

Author Response

It would have be nice to see some theoretical runtime model, that on
the one hand side predicts/models the actual runtime on the machine, and in the other hand side gives more insight into the properties both of the algorithm and of the system, such that the experiments could be exported to other applications (both in the context of algorithms as well as in the context of using a specific GPU/CPU architecture).

A simple extension could be: just use some different GPU (and/or CPU) and analyse what happens there.

What are the reasons for the fact that HA4 is affected at 60% (percolation threshold) but TMK not? Well, actually it is affected looking at Fig.3, but in a positive sense!

Thank you for the suggestions. We have extended the discussion of the results (see bold text on page 8) to give more insights into the properties of the algorithm. Further, we have repeated the benchmarks on a different PC and added the results in Figure 5. Since the results of the two PCs where qualitatively very similar to one another (regarding relative performance to the HA4 algorithm as well as kernel fractions), we have not included them in Figures 4 and 6.

Round 2

Reviewer 1 Report

The authors have addressed all of my concerns with the original manuscript. The revised manuscript is ready for publication.

Reviewer 2 Report

Thank you for taking my comments into account.